# On sensitivity of meta-learning to support data

**Mayank Agarwal** [1]
mayank.agarwal@ibm.com

**Mikhail Yurochkin** [1,2]
mikhail.yurochkin@ibm.com

**Yuekai Sun** [3]
yuekai@umich.edu

IBM Research,[1] MIT-IBM Watson AI Lab,[2] University of Michigan[3].

## Abstract

Meta-learning algorithms are widely used for few-shot learning. For example, image recognition systems that readily adapt to unseen classes after seeing only a few labeled examples. Despite their success, we show that modern meta-learning algorithms are extremely sensitive to the data used for adaptation, i.e. support data. In particular, we demonstrate the existence of (unaltered, in-distribution, natural) images that, when used for adaptation, yield accuracy as low as 4% or as high as 95% on standard few-shot image classification benchmarks. We explain our empirical findings in terms of class margins, which in turn suggests that robust and safe meta-learning requires larger margins than supervised learning.

## 1   Introduction

Meta-learning, or learning to learn [29], is the problem of training models that can adapt to new tasks quickly, using only a handful of examples. The problem is inspired by humans' ability to learn new skills or concepts at a rapid pace (*e.g.* recognizing previously unknown objects after seeing only a single example [20]). Meta-learning has found applications in many domains, including safety-critical medical image analysis [23], autonomous driving [35], visual navigation [43] and legged robots control [40]. In this paper, we investigate the vulnerabilities of modern meta-learning algorithms in the context of few-shot image classification problem [25], where a meta-learner needs to solve a classification task on classes unseen during training using a small number (typically 1 or 5) of labeled samples from each of these classes to adapt. Specifically, we demonstrate that the performance of modern meta-learning algorithms on few-shot image recognition benchmarks varies drastically depending on the examples provided for adaptation, typically called *support data*, raising concerns about its safety in deployment.

Despite the many empirical successes of artificial intelligence, its vulnerabilities are important to explore and mitigate in our pursuit of safe AI that is suitable for critical applications. Sensitivity to small (possibly adversarial) perturbations to the inputs [13], backdoor attacks allowing malicious model manipulations [6], algorithmic biases [2] and poor generalization outside of the training domain [18] are some of the prominent examples of AI safety failures. Some of these issues have also been studied in the context of meta-learning, e.g. adversarial robustness [45, 11, 44] and fairness [37]. Most of the aforementioned AI-safety dangers are associated with adversarial manipulations of the train or test data, or significant distribution shifts at test time. In meta-learning, prior works demonstrated that an adversary can create visually imperceptible changes of the test inputs [11] or the support data [44, 30] causing meta-learning algorithms to fail on various few-shot image recognition benchmarks. In this work we demonstrate *adversary-free* and *in-distribution* failures specific to meta-learning. Sohn et al. [39] studied a similar problem in the context of semi-supervised learning.

A distinct feature of meta-learning is the adaptation stage where the model is updated using a scarce amount of labeled examples from a new task. In practice, these examples need to be selected and presented to a human for labeling. Then a meta-learner adapts and can be used to classify the

35th Conference on Neural Information Processing Systems (NeurIPS 2021).

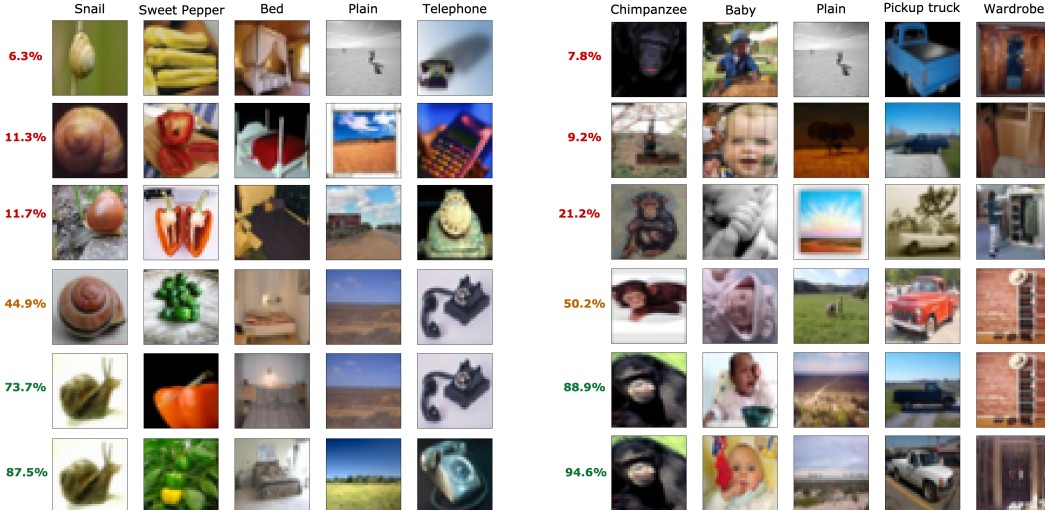

Figure 1: Examples of CIFAR-FS support images (unaltered, i.e. not modified adversarially or in any other way) from two test tasks that yield vastly different 1-shot post-adaptation performances of a popular meta-learning algorithm (MetaOptNet-SVM). Accuracy ranges from 6.3% to 94.6% suggesting extreme sensitivity of the meta-learner to the support data. Images mostly appear representative of the corresponding classes and would be hard to recognize as potentially problematic without significant expertise of the training dataset from the human labeling the support data.

remaining images without a human. Currently, meta-learning algorithms choose support examples for labeling at random. Dhillon et al. [8] noted that the standard deviation of accuracies computed with random support examples may be large. We demonstrate that if the support data is not carefully curated, the performance of the meta-learner can be unreliable. In Figure 1 we present examples of unaltered support images (i.e. they are not modified adversarially or in any other way) representative of the corresponding task (in-distribution) that lead to vastly different performance of the meta-learner after adaptation. Support examples causing poor performance are not prevalent, however, they are also not data artifacts as there are multiple of them. The existence of such examples might be concerning when deploying meta-learning in practice, even when adversarial intervention is out of scope.

Our main contributions are as follows:

1. We present a simple algorithm for finding the best and the worst support examples and empirically demonstrate the sensitivity of popular meta-learning algorithms to support data.

2. We demonstrate that a popular strategy for achieving robustness [22] adapted to our setting *fails* to solve the support data sensitivity problem.

3. We explain the existence of the worst case examples from the margin perspective suggesting that robust meta-learning requires class margins more stringent than classical supervised learning.

## 2 Meta-learning approaches

Meta-learning approaches are typically categorized into model-based [34, 26, 33], metric-based [17, 41, 38] and optimization-based methods [10, 28, 14]. We refer to Hospedales et al. [15] for a recent survey of the area. In this paper we mostly focus on the optimization-based method popularized by the Model Agnostic Meta Learning (MAML) [10] bi-level formulation (metric-based prototypical networks [38] are also considered in the experiments). Let $\{\mathcal{T}_n\}_{i=1}^n$ be a collection of $n$ tasks $\mathcal{T}_i = \{\mathcal{A}_i, \mathcal{D}_i\}$ each consisting of a support $\mathcal{A}_i$ and a query (or validation) $\mathcal{D}_i$ datasets. MAML

bi-level optimization problem is as follows [10]:

$$\min_\theta \frac{1}{n} \sum_{i=1}^n \ell(\theta_i, \theta; \mathcal{D}_i),$$
$$\text{such that } \theta_i = \arg\min_{\theta_i} \ell(\theta_i, \theta; \mathcal{A}_i), \; i = 1, \ldots, n. \tag{2.1}$$

The first line of equation 2.1 is typically minimized using gradient-based optimization of $\theta$ and is called *meta-update*. The second line is the *adaptation* step and its implementation differs depending on whether the fine-tuned $\theta_i$ are treated as parameters of the end-to-end neural network or as the "head" parameters, i.e. the last linear classification layer. We refer to Goldblum et al. [12] for an empirical study of the differences between the two adaptation perspectives.

MAML [10] and Reptile [28] are examples of algorithms that fine-tune all network parameters during the adaptation. Instead of solving the argmin of $\theta_i$ exactly they approximate it with a small number of gradient steps. For one gradient step approximation equation 2.1 can be written as

$$\min_\theta \frac{1}{n} \sum_{i=1}^n \ell(\theta - \alpha \frac{\partial \ell(\theta; \mathcal{A}_i)}{\partial \theta}; \mathcal{D}_i) \tag{2.2}$$

for some step size $\alpha$. MAML directly optimizes equation 2.2 differentiating through the inner gradient step, which requires expensive Hessian computations. Reptile introduces an approximation by-passing the Hessian computations and often performs better.

Another family of meta-learning algorithms considers $\theta$ as the neural network feature extractor parameters shared across tasks and adapts only the linear classifier parametrized with $\theta_i$, that takes the features extracted with $\theta$ as inputs. The advantage of this perspective is that the adaptation minimization problem is convex and, for many linear classifiers, can be solved fast and exactly. Let $a(\theta, \mathcal{A})$ denote a procedure that takes data in $\mathcal{A}$, passes it through a feature extractor parametrized with $\theta$, and returns $\theta_i$, i.e. optimal parameters of a linear classifier using the obtained features to predict the corresponding labels. Then equation 2.1 can be written as

$$\min_\theta \frac{1}{n} \sum_{i=1}^n \ell(\theta, a(\theta, \mathcal{A}_i); \mathcal{D}_i). \tag{2.3}$$

This approach requires $a(\theta, \mathcal{A})$ to be differentiable. R2D2 [4] casts classification as a multi-target ridge-regression problem and utilizes the corresponding closed-form solution as $a(\theta, \mathcal{A})$. Lee et al. [21] propose MetaOptNet, where $a(\theta, \mathcal{A})$ is a differentiable quadratic programming solver [1], and implement it with linear support vector machines (SVM) and ridge-regression. Prototypical networks [38] is a metric-based approach that can also be viewed from the perspective of equation 2.3. Here $a(\theta, \mathcal{A})$ outputs class centroids in the neural-feature space and the predictions are based on the closest class centroids. Last-layer adaptation approaches, especially MetaOptNet, outperform full-network approaches on most benchmarks [21, 12].

Both adaptation perspectives have a weakness underlying our study: their test performance is based on an optimization problem with large number of parameters and as little as 5 data points (1-shot 5-way setting). To understand the problem, consider the last-layer adaptation approach: even when the feature extractor produces linearly separable representations, the dimension is large (for example, Lee et al. [21] utilize a ResNet-12 architecture with 2560-dimensional last layer features that we adopt in our experiments) making the corresponding linear classifier extremely sensitive to the support data. In Section 3 we demonstrate the problem empirically and provide theoretical insights in Section 4.

## 2.1 Meta-learning benchmarks

Before presenting our findings, we discuss the meta-learning benchmarks we consider. Meta-learning algorithms are often compared in a few-shot image recognition setting. Each task typically has five unique classes, i.e. 5-way, and one or five examples per class for adaptation, i.e. 1-shot or 5-shot. Classes that appear at test time are not seen during training. Few-shot learning datasets are typically the derivatives of the existing supervised learning benchmarks, e.g. CIFAR-100 [19] and ImageNet [7]. Few-shot problem is setup by disjoint partitioning of the available classes into train, validation and test. Tasks are obtained by sampling 5 distinct classes from the corresponding pool and assigning

a subset of examples for adaptation and a different subset for meta-update during training or accuracy evaluation for reporting the performance.

CIFAR-FS [4] is a dataset of 60000 32×32 RGB images from CIFAR-100 partitioned into 64, 16 and 20 classes for training, validation and testing, respectively. FC-100 [31] is also a derivative of CIFAR-100 with a different partition aimed to reduce semantic overlap between 60 classes assigned for training, 20 for validation, and 20 for testing. MiniImageNet [41] is a subsampled, downsized version of ImageNet. It consists of 60000 84×84 RGB images from 100 classes split into 64 for training, 16 for validation, and 20 for testing.

## 3 Finding adaptation vulnerabilities

In this section we study the performance range of meta-learners trained with a variety of algorithms: MAML [10], Meta-Curvature (MC) [32], Prototypical networks [38], R2D2 [4], and MetaOptNet with Ridge and SVM heads [21]. At test time a meta-learner receives support data and its performance is recorded after it adapts. Typical evaluation protocols sample support data randomly and report average performance across tasks. While this is a useful measure for comparing algorithms, safety-critical applications demand understanding of potential performance ranges. We demonstrate that there are support examples the lead to vastly different test performances. To identify the worst and the best case support examples for a given task we perform an iterative greedy search.

Let $\mathcal{X} = \{x_m^k\}_{m \in [M]}^{k \in [K]}$ be a set of potential support examples for a given task. $K$ is the number of classes, $M$ is the number of examples per class (we assume same $M$ for each class for simplicity), and $[M] = \{0, \dots, M-1\}$. Let $\mathcal{D}$ be the evaluation set for this task. In a $J$-shot setting, let $\mathcal{Z} = \{z_j^k\}_{j \in [J]}^{k \in [K]}$ be a set of *indices* of the support examples ($z_j^k \neq z_{j'}^k$ for any $k, j \neq j'$ and $z_j^k \in [M]$ for any $k, j$). Denote $R(\mathcal{Z}, \mathcal{X}, \mathcal{D})$ a function computing accuracy of a meta learner on the query set $\mathcal{D}$ after adapting on $\mathcal{A} = \{x_{z_j^k}^k\}_{j \in [J]}^{k \in [K]}$. Finding the worst/best case accuracy amounts to finding indices $\mathcal{Z}$ minimizing/maximizing $R(\mathcal{Z}, \mathcal{X}, \mathcal{D})$. We solve this greedily by updating a single index $z_j^k$ at a time, holding the rest of $\mathcal{Z}$ fixed, iterating over $[K]$ and $[J]$ multiple times. We summarize the procedure for finding the worst case accuracy in Algorithm 1 (the best case accuracy is analogous).[1] Due to the greedy nature, this algorithm finds a local optima, but it is sufficiently effective as we see in the following section.

---

**Algorithm 1** Finding the worst case support examples

---

**Input:** trained meta-learner, potential support examples $\mathcal{X}$ to search over, query data $\mathcal{D}$.
**repeat**
    Initialize indices $\mathcal{Z} = \{z_j^k\}_{j \in [J]}^{k \in [K]}$ randomly.
    **for** $j \in [J]$, $k \in [K]$ **do**
        $z_j^k \leftarrow \arg\min_{z_j^k} R(\mathcal{Z}, \mathcal{X}, \mathcal{D})$, $z_j^k \in [M] \setminus \{z_{j'}^k\}_{j' \neq j}$
    **end for**
**until** $R(\mathcal{Z}, \mathcal{X}, \mathcal{D})$ stops decreasing
**Output:** support examples indices $\mathcal{Z} = \{z_j^k\}_{j \in [J]}^{k \in [K]}$

---

**Visualizing the worst case support search**    In Figure 2 we visualize a single iteration of Algorithm 1 on a 1-shot task from the CIFAR-FS dataset. This is a 5-way task with "snail", "red pepper", "bed", "plain", and "telephone" as classes. In round 1, i.e. $k = 0$, of iteration 1 we start with a random example per class and evaluate post-adaptation accuracy on the query data $\mathcal{D}$ for each potential support "snail", i.e. $\{x_m^0\}_{m \in [M]}$. Here the first line corresponds to "snail" indexed with $m = 0$, i.e. $x_0^0$, and post-adaptation accuracy of 73.5%. We select a "snail" support image corresponding to the worst accuracy of 53.8% and proceed to round 2, i.e. $k = 1$, of iteration 1. In round 2 we repeat the same procedure for the corresponding class "red pepper" (again the first line corresponds to "red pepper" indexed with $m = 0$, i.e. $x_0^1$), holding the support "snail" image selected previously and

---

[1]This is a simple approach, however we found it faster and more efficient then a more sophisticated continuous relaxation using weights of the potential support examples with sparsity constraints.

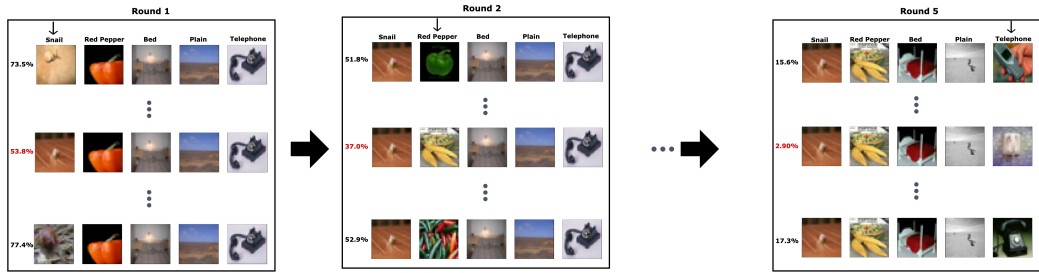

Figure 2: Visualization of Algorithm 1 on a random 1-shot task for the MetaOptNet-SVM method on the CIFAR-FS dataset. We show 1 iteration of the algorithm here, where the algorithm starts by randomly sampling a support image per class and then iterates over all data samples for all classes.

support images for other classes fixed. The algorithm finds the worst case support "red pepper" image corresponding to the post-adaptation accuracy of 37%. Then the algorithm proceeds analogously to rounds 3, 4, and 5 of iteration 1, resulting in a combination of support examples corresponding to 2.9% post-adaptation accuracy. This is already sufficiently low, however on some tasks and higher-shot settings it may be beneficial to run additional iterations. On iteration 2, the algorithm will again go through all the classes starting from the support examples found on iteration 1 instead of random ones. In our experiments we always run Algorithm 1 for 3 iterations. In Appendix B we empirically study the convergence of the algorithm justifying this choice.

Table 1: Accuracies for different meta-learning methods on the CIFAR-FS dataset

|  |  | Worst acc | Avg acc | Best acc |
|---|---|---|---|---|
| MAML | 1-shot | $5.91 \pm 1.94\%$ | $56.52 \pm 10.85\%$ | $80.04 \pm 6.83\%$ |
|  | 5-shot | $13.88 \pm 7.28\%$ | $70.45 \pm 8.42\%$ | $85.15 \pm 6.12\%$ |
|  | 10-shot | $26.26 \pm 7.85\%$ | $70.88 \pm 7.82\%$ | $85.81 \pm 6.47\%$ |
| MC | 1-shot | $4.73 \pm 1.51\%$ | $47.76 \pm 11.28\%$ | $69.81 \pm 8.37\%$ |
|  | 5-shot | $9.29 \pm 6.13\%$ | $70.23 \pm 8.82\%$ | $85.22 \pm 6.53\%$ |
|  | 10-shot | $16.95 \pm 10.78\%$ | $70.62 \pm 7.89\%$ | $87.81 \pm 6.91\%$ |
| ProtoNets | 1-shot | $5.06 \pm 2.37\%$ | $63.80 \pm 0.71\%$ | $84.88 \pm 6.29\%$ |
|  | 5-shot | $18.28 \pm 9.85\%$ | $80.06 \pm 0.46\%$ | $90.41 \pm 4.72\%$ |
|  | 10-shot | $27.32 \pm 11.17\%$ | $82.95 \pm 0.44\%$ | $91.44 \pm 4.35\%$ |
| R2D2 | 1-shot | $6.14 \pm 2.89\%$ | $68.86 \pm 0.68\%$ | $86.63 \pm 5.97\%$ |
|  | 5-shot | $14.73 \pm 8.21\%$ | $82.29 \pm 0.44\%$ | $92.34 \pm 4.21\%$ |
|  | 10-shot | $29.30 \pm 12.16\%$ | $85.74 \pm 0.42\%$ | $93.41 \pm 3.81\%$ |
| MetaOptNet-Ridge | 1-shot | $5.17 \pm 2.97\%$ | $71.21 \pm 0.67\%$ | $87.40 \pm 5.95\%$ |
|  | 5-shot | $16.81 \pm 11.57\%$ | $84.18 \pm 0.45\%$ | $93.15 \pm 4.43\%$ |
|  | 10-shot | $32.10 \pm 16.26\%$ | $86.82 \pm 0.42\%$ | $93.89 \pm 3.59\%$ |
| MetaOptNet-SVM | 1-shot | $5.27 \pm 2.82\%$ | $70.79 \pm 0.69\%$ | $87.65 \pm 5.76\%$ |
|  | 5-shot | $14.92 \pm 8.60\%$ | $83.98 \pm 0.44\%$ | $93.36 \pm 4.60\%$ |
|  | 10-shot | $22.24 \pm 10.13\%$ | $87.11 \pm 0.40\%$ | $93.56 \pm 3.98\%$ |

### 3.1 Performance range results

We summarize the worst, average and best accuracies of six meta-learning algorithms on three benchmark datasets (see Section 2.1 for data descriptions) in 1-shot, 5-shot, and 10-shot setting in Tables 1, 2, and 3. All meta-learners are trained using code from the authors or more modern meta-learning libraries [3] (see Appendix A for implementation and additional experimental details). For evaluation we randomly partition each class in each task into 400 potential support examples composing $\mathcal{X}$ and 200 query examples composing $\mathcal{D}$ (all datasets have 600 examples per class). To compute average accuracy we randomly sample corresponding number of support examples per class; for the best and the worst case accuracies we use Algorithm 1 to search over support examples in $\mathcal{X}$.

Table 2: Accuracies for different meta-learning methods on the FC100 dataset

|  |  | Worst acc | Avg acc | Best acc |
|---|---|---|---|---|
| MAML | 1-shot | $7.32 \pm 1.49\%$ | $31.89 \pm 6.75\%$ | $50.13 \pm 6.75\%$ |
|  | 5-shot | $7.51 \pm 3.59\%$ | $43.58 \pm 7.61\%$ | $61.64 \pm 8.50\%$ |
|  | 10-shot | $10.51 \pm 2.86\%$ | $44.30 \pm 6.89\%$ | $64.98 \pm 7.27\%$ |
| MC | 1-shot | $6.53 \pm 1.56\%$ | $36.56 \pm 8.05\%$ | $56.75 \pm 5.57\%$ |
|  | 5-shot | $5.17 \pm 1.34\%$ | $47.12 \pm 7.02\%$ | $66.33 \pm 5.11\%$ |
|  | 10-shot | $8.35 \pm 3.85\%$ | $49.12 \pm 6.67\%$ | $65.27 \pm 5.58\%$ |
| ProtoNets | 1-shot | $5.25 \pm 1.69\%$ | $37.21 \pm 0.50\%$ | $59.75 \pm 6.39\%$ |
|  | 5-shot | $5.57 \pm 2.99\%$ | $50.49 \pm 0.48\%$ | $70.31 \pm 6.41\%$ |
|  | 10-shot | $9.93 \pm 4.39\%$ | $56.15 \pm 0.47\%$ | $72.94 \pm 6.23\%$ |
| R2D2 | 1-shot | $6.13 \pm 1.63\%$ | $37.91 \pm 0.48\%$ | $59.67 \pm 6.17\%$ |
|  | 5-shot | $6.72 \pm 2.85\%$ | $54.35 \pm 0.49\%$ | $74.34 \pm 6.59\%$ |
|  | 10-shot | $12.00 \pm 7.07\%$ | $61.72 \pm 0.47\%$ | $77.94 \pm 3.56\%$ |
| MetaOptNet-Ridge | 1-shot | $5.44 \pm 1.57\%$ | $39.13 \pm 0.51\%$ | $61.28 \pm 6.18\%$ |
|  | 5-shot | $5.97 \pm 3.29\%$ | $53.20 \pm 0.47\%$ | $72.65 \pm 6.43\%$ |
|  | 10-shot | $11.56 \pm 2.78\%$ | $59.52 \pm 0.48\%$ | $75.30 \pm 1.56\%$ |
| MetaOptNet-SVM | 1-shot | $5.29 \pm 1.57\%$ | $38.19 \pm 0.48\%$ | $60.14 \pm 6.12\%$ |
|  | 5-shot | $5.75 \pm 2.83\%$ | $54.45 \pm 0.49\%$ | $74.01 \pm 6.64\%$ |
|  | 10-shot | $9.54 \pm 3.86\%$ | $60.52 \pm 0.48\%$ | $77.03 \pm 6.27\%$ |

Table 3: Accuracies for different meta-learning methods on the miniImageNet dataset

|  |  | Worst acc | Avg acc | Best acc |
|---|---|---|---|---|
| MAML | 1-shot | $6.08 \pm 1.77\%$ | $47.13 \pm 8.78\%$ | $71.39 \pm 6.74\%$ |
|  | 5-shot | $10.15 \pm 8.40\%$ | $57.69 \pm 7.92\%$ | $79.60 \pm 5.43\%$ |
|  | 10-shot | $20.88 \pm 7.22\%$ | $59.52 \pm 8.34\%$ | $79.94 \pm 3.41\%$ |
| MC | 1-shot | $4.46 \pm 2.05\%$ | $45.03 \pm 8.79\%$ | $65.98 \pm 6.23\%$ |
|  | 5-shot | $5.79 \pm 3.45\%$ | $60.47 \pm 7.57\%$ | $75.09 \pm 4.72\%$ |
|  | 10-shot | $9.67 \pm 3.84\%$ | $60.54 \pm 7.45\%$ | $73.23 \pm 6.24\%$ |
| ProtoNets | 1-shot | $4.69 \pm 2.16\%$ | $53.42 \pm 0.59\%$ | $76.46 \pm 5.64\%$ |
|  | 5-shot | $9.53 \pm 4.91\%$ | $70.60 \pm 0.43\%$ | $85.33 \pm 3.60\%$ |
|  | 10-shot | $15.39 \pm 5.17\%$ | $75.91 \pm 0.38\%$ | $87.31 \pm 3.24\%$ |
| R2D2 | 1-shot | $6.10 \pm 3.27\%$ | $56.09 \pm 0.58\%$ | $78.17 \pm 5.33\%$ |
|  | 5-shot | $12.03 \pm 5.51\%$ | $72.04 \pm 0.43\%$ | $86.64 \pm 3.34\%$ |
|  | 10-shot | $15.78 \pm 6.10\%$ | $77.32 \pm 0.36\%$ | $86.70 \pm 1.99\%$ |
| MetaOptNet-Ridge | 1-shot | $5.19 \pm 3.21\%$ | $57.94 \pm 0.62\%$ | $79.15 \pm 5.05\%$ |
|  | 5-shot | $9.83 \pm 4.05\%$ | $74.80 \pm 0.43\%$ | $84.81 \pm 4.12\%$ |
|  | 10-shot | $19.16 \pm 10.07\$$ | $80.31 \pm 0.36\%$ | $89.72 \pm 2.99\%$ |
| MetaOptNet-SVM | 1-shot | $4.82 \pm 3.03\%$ | $59.03 \pm 0.62\%$ | $80.38 \pm 5.40\%$ |
|  | 5-shot | $9.52 \pm 4.88\%$ | $75.54 \pm 0.40\%$ | $85.41 \pm 3.86\%$ |
|  | 10-shot | $15.93 \pm 7.64\%$ | $80.16 \pm 0.37\%$ | $90.63 \pm 2.71\%$ |

Our key finding is the large range of performances of *all* meta-learning algorithms considered in 1-, 5-, and 10-shot settings. Prototypical networks have slightly better worst-case 5-shot accuracy on CIFAR-FS, but it has large variance and is likely due to our algorithm finding poor local optima on one of the tasks. We also see no differences between meta-learners adapting end-to-end, i.e. MAML and MC, and those adapting only the last linear classification layer, i.e. R2D2 and MetaOptNet. Goldblum et al. [12] showed that the last-layer adaptation methods produce good quality linear-separable embeddings. One could expect such methods to be less sensitive to support data, however, as we discuss in Section 4, linear separability is not sufficient in the few-shot learning setting. 10-shot worst-case accuracies are not as poor (despite mostly remaining worse than random guessing), but

we expect that with more support data available, the gap would narrow.[2] Finally, we note that the best-case accuracy is significantly better, especially in the 1-shot setting.

## 3.2 Worst case support examples are not artifacts

We have demonstrated that it is possible to find support examples yielding poor post-adaptation performance of a variety of meta-learners. It is also important to understand the nature of these worst-case examples: are they data artifacts, i.e. outliers or miss-labeled examples, or realistic images that could cause failures in practice? We argue that the latter is the case.

1. In Figure 1 we presented several examples of the worst-case support images on CIFAR-FS: they are correctly labeled and appear representative of the respective classes. Inspecting the images on the left closer we note that it is often not easy to notice visually that such support examples could result in a poor performance without significant expert knowledge of the dataset. Only the cellphone image labelled "telephone" and grey-scale image labeled "plain" seem potentially problematic. On the other hand, "snail" and "bed" images all appear normal. On the right figure, majority of the images also appear reasonable.

2. 10-shot setting should be a lot more resilient to outliers, however Algorithm 1 continues to be successful in 10-shot setting, finding ten different examples per class leading to accuracy slightly better than a random predictor as shown in Tables 1, 2, and 3.

3. In Figure 3 we present histograms of accuracies visualizing the first iteration over classes of Algorithm 1 in 1-shot learning on CIFAR-FS. The right most histogram corresponds to post-adaptation accuracies for different choices of support image for class 0 and random choices for classes 1-4. The subsequent histogram is for different choices of support images for class 1, where image for class 0 is chosen with Algorithm 1 and classes 2-4 are random, and analogously for the remaining three histograms. We see that the lower accuracy tails of the first two histograms contain multiple worst-case support examples in the corresponding classes. By the third histogram, the range of accuracies is below 50% for *all* possible support examples in the corresponding classes.

4. In Figure 4 we present histogram of accuracies of 3991 unique combinations from CIFAR-FS of 1-shot support examples evaluated by Algorithm 1 throughout 3 iterations. There are 3335 distinct sets of 5 examples each with less than 50% post-adaptation accuracy.

We present analogous analysis for other meta-learners and datasets in Appendix C, where we also conclude that worst-case adaptation examples are realistic and can cause malfunctions in practice.

## 3.3 Improving support data robustness with adversarial training

The issue of robustness has been studied in many contexts in machine learning, and adversarial vulnerability of deep learning based models has been explored extensively in the recent literature [13, 5, 22]. While the support data sensitivity of meta learners presented in this work is a new type of non-robustness, it is possible to approach the problem borrowing ideas from adversarial training [22]. The high-level idea of adversarial training is to use a mechanism exposing non-robustness to guide the data used for model updates. For example, if a model is sensitive to perturbations in the inputs, for an incoming batch of data we find the worst-case (by maximizing the loss) perturbations to the inputs and use the perturbed inputs to updated model parameters. To converge, adversarial training needs to find model parameters such that the mechanism exposing non-robustness can no-longer damage the performance. Adversarial training has theoretical guarantees for convex models [42] and has been shown empirically to be successful in defending against adversarial attacks on deep learning models [22].

We use adversarial training scheme in an attempt to achieve robustness to support data in meta learning. Specifically, we use Algorithm 1 to find worst-case adaptation examples during training instead of using random ones as in standard training. The result is quite intriguing: Tables 4 and 5 summarize performance of adversarially (in a sense of support data) trained meta-learners on train and test tasks. In most cases, adversarial training converged, i.e. we are no longer able to find detrimental worst-case support examples with Algorithm 1 on the *training* tasks, however we observe

---

[2] With enough support data, last-layer methods should succeed as their embeddings are linearly separable.

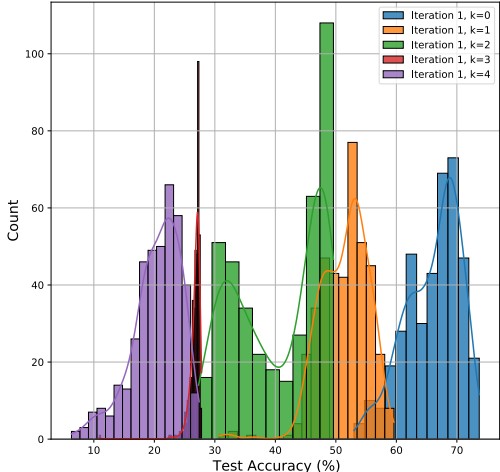

Figure 3: Histogram of test accuracies on the first iteration of Algorithm 1 as it progresses through classes evaluating 1-shot combinations of images for adaptation on a CIFAR-FS test task with MetaOptNet-SVM meta-learner.

Figure 4: Histogram of test accuracies computed during 3 iterations of Algorithm 1 evaluating different unique 1-shot combinations of images for adaptation on a CIFAR-FS test task with MetaOptNet-SVM meta-learner.

no improvements of the worst-case accuracy on the *test* tasks. Our experiment demonstrates that support data sensitivity in meta-learning is not easily addressed with existing methods and requires exploring new solution paths.

Table 4: Accuracies for different meta-learning methods trained in the standard manner and adversarially on the CIFAR-FS dataset

| Method | Dataset | Training | Worst acc | Avg acc | Best acc |
|---|---|---|---|---|---|
| R2D2 | Train | Standard | $13.83 \pm 9.35$ % | $87.68 \pm 0.56\%$ | $96.91 \pm 3.15\%$ |
| | | Adversarial | $46.34 \pm 14.93\%$ | $88.19 \pm 0.59\%$ | $97.94 \pm 2.89\%$ |
| | Test | Standard | $6.14 \pm 2.89\%$ | $68.86 \pm 0.68\%$ | $86.63 \pm 5.97\%$ |
| | | Adversarial | $6.76 \pm 2.69\%$ | $68.62 \pm 0.66\%$ | $87.46 \pm 5.86\%$ |
| MetaOptNet-Ridge | Train | Standard | $77.75 \pm 17.80\%$ | $99.23 \pm 0.15\%$ | $99.81 \pm 0.99\%$ |
| | | Adversarial | $90.08 \pm 14.22\%$ | $98.87 \pm 0.21\%$ | $99.84 \pm 0.66\%$ |
| | Test | Standard | $5.17 \pm 2.96\%$ | $71.21 \pm 0.67\%$ | $87.41 \pm 5.95\%$ |
| | | Adversarial | $5.38 \pm 2.79\%$ | $71.81 \pm 0.67\%$ | $88.42 \pm 5.60\%$ |
| MetaOptNet-SVM | Train | Standard | $9.74 \pm 9.00\%$ | $91.93 \pm 0.47\%$ | $97.58 \pm 2.51\%$ |
| | | Adversarial | $93.47 \pm 11.84\%$ | $99.08 \pm 0.19\%$ | $99.81 \pm 0.89\%$ |
| | Test | Standard | $5.27 \pm 2.82\%$ | $70.79 \pm 0.69\%$ | $87.66 \pm 5.76\%$ |
| | | Adversarial | $4.97 \pm 2.58\%$ | $71.11 \pm 0.70\%$ | $87.84 \pm 5.93\%$ |

## 4 Meta-learning margin analysis

Meta-learners, specifically those only adapting the last layer linear classifier, produce embeddings that appear (approximately) linearly separable even when projected into two dimensions as shown in Figure 5 (we used Multidimensional Scaling [24] to preserve the relative cluster sizes and distances between clusters). In the supervised learning context this would be considered an easy problem for, e.g., linear SVM as in the MetaOptNet-SVM meta-learner. The problem, however, is that in the supervised learning context we typically have sufficient data, while in the few-shot setting meta-learners are restricted to as little as a single example per class in a high-dimensional space.

Lets take another look at Figure 5. These are Multidimensional Scaling [24] projections of the embeddings obtained with the MetaOptNet trained on CIFAR-FS. Embeddings in (a) and (c) correspond

Table 5: Accuracies for different meta-learning methods trained in the standard manner and adversarially on the FC100 dataset

| Method | Dataset | Training | Worst acc | Avg acc | Best acc |
|---|---|---|---|---|---|
| R2D2 | Train | Standard | $8.95 \pm 6.62\%$ | $84.24 \pm 0.60\%$ | $95.48 \pm 3.45\%$ |
| | | Adversarial | $45.01 \pm 12.55\%$ | $87.50 \pm 0.57\%$ | $97.11 \pm 2.97\%$ |
| | Test | Standard | $6.13 \pm 1.63\%$ | $37.91 \pm 0.48\%$ | $59.67 \pm 6.17\%$ |
| | | Adversarial | $6.44 \pm 1.68\%$ | $38.70 \pm 0.46\%$ | $60.87 \pm 6.30\%$ |
| MetaOptNet-Ridge | Train | Standard | $14.27 \pm 11.32\%$ | $92.54 \pm 0.45\%$ | $97.62 \pm 2.51\%$ |
| | | Adversarial | $79.67 \pm 14.69\%$ | $96.83 \pm 0.38\%$ | $98.76 \pm 2.26\%$ |
| | Test | Standard | $5.44 \pm 1.57\%$ | $39.13 \pm 0.51\%$ | $61.29 \pm 6.18\%$ |
| | | Adversarial | $5.47 \pm 1.89\%$ | $36.87 \pm 0.48\%$ | $59.07 \pm 6.63\%$ |
| MetaOptNet-SVM | Train | Standard | $18.05 \pm 12.64\%$ | $94.13 \pm 0.38\%$ | $98.28 \pm 2.09\%$ |
| | | Adversarial | $90.37 \pm 12.18\%$ | $97.12 \pm 0.39\%$ | $99.06 \pm 1.98\%$ |
| | Test | Standard | $5.29 \pm 1.57\%$ | $38.19 \pm 0.48\%$ | $60.14 \pm 6.11\%$ |
| | | Adversarial | $6.08 \pm 1.78\%$ | $35.92 \pm 0.45\%$ | $60.30 \pm 6.26\%$ |

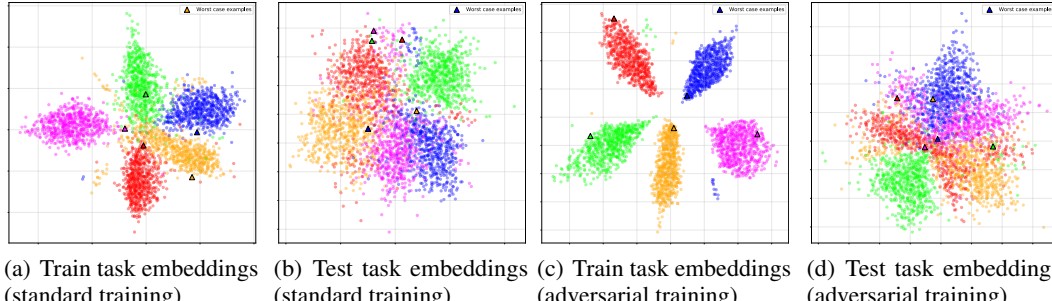

(a) Train task embeddings (standard training)  (b) Test task embeddings (standard training)  (c) Train task embeddings (adversarial training)  (d) Test task embeddings (adversarial training)

Figure 5: Projected embeddings of MetaOptNet-SVM for a train and a test task query data from CIFAR-FS. We compare standard and adversarial training discussed in Section 3.3. Points are colored with their labels. Highlighted points are the worst-case support examples selected with Algorithm 1.

to a query data from a train task for standard and adversarially trained models, and embeddings in (b) and (d) to a query data from a test task for the corresponding models. All embeddings appear well-clustered, however embeddings in (c) have the largest separation and the smallest within-class variance. This wouldn't make a significant difference in supervised learning with enough data, but makes a big difference for meta-learning: when using Algorithm 1 to find the worst-case support examples (highlighted in the figures), the corresponding accuracies are 1.8% for (a), 2.90% for (b), 99.4% for (c), and 5.70% for (d). We see that although all embeddings are well-separated[3], only (c) is robust to support data selection. We present the projected embeddings for a variety of meta-learners on the FC100 dataset in Appendix D. As we discuss next, robust meta-learning, in addition to vanilla linear-separability, requires features with bigger class separation and lower intra-class variances.

In the following theorem, we show that as long as the class embeddings are sufficiently separated, the probability of any two points sampled from the classes leading to a max-margin classifier with large misclassification rate is exponentially small. We note the high degree of separation (linear in embedding dimension) necessary to guarantee robustness to the choice of support data. This suggests unless the class embeddings are well-separated, the resulting meta-learning algorithm will be sensitive to the choice of support data.

**Theorem 4.1.** *Consider a (binary) Gaussian discriminant analysis model:*

$$X \mid Y = 1 \sim N(\mu, \sigma^2 I_d),$$
$$X \mid Y = 0 \sim N(-\mu, \sigma^2 I_d).$$

---

[3]For comparison, the accuracies of the corresponding supervised learning problems (i.e. linear classifiers trained using embeddings of all 400 per class potential support examples $\mathcal{X}$, rather than a single example per class) are 99.6% for (a) and 94.3% for (b).

*As long as $\mu = (d + \sqrt{2dt} + 2t)\sigma$ for some $t > 0$, then the max-margin classifier between two points sampled independently from each cluster has misclassification rate at most $\Phi(-d - \sqrt{2dt} - 2t)$ with probability at least $(1 - e^{-t})^2$.*

*Proof.* Consider a (binary) Gaussian discriminant analysis model:

$$X \mid Y = 1 \sim N(\mu, \sigma^2 I_d),$$
$$X \mid Y = 0 \sim N(-\mu, \sigma^2 I_d).$$

Define the core of the clusters as the sets

$$\mathcal{C}_1 \triangleq \{x \in \mathbb{R}^d \mid \|x - \mu\|_2 \leq r\sigma\},$$
$$\mathcal{C}_0 \triangleq \{x \in \mathbb{R}^d \mid \|x + \mu\|_2 \leq r\sigma\}.$$

It is a tedious geometric exercise to show that as long as $\mu > 2r\sigma$, then the midpoint of any pair of points $(x_1, x_0) \in \mathcal{C}_1 \times \mathcal{C}_0$ falls outside $\mathcal{C}_1 \cup \mathcal{C}_0$. Further, the hyperplane

$$\mathcal{H} \triangleq \{x \in \mathbb{R}^d \mid (x_1 - x_0)^T x = \tfrac{1}{2}(\|x_1\|_2^2 - \|x_0\|_2^2)\}$$

bisects $\mathcal{C}_0$ and $\mathcal{C}_1$ for any choice of $x_1, x_0$. This implies the risk of any max-margin classifier constructed from $x_1$ and $x_0$ has misclassification error rate at most $2\Phi(-r)$, where $\Phi$ is the $N(0, 1)$ CDF. We note that the probability of a pair of independently sampled points $x_1 \sim N(\mu, \sigma^2 I_d)$ and $x_0 \sim N(-\mu, \sigma^2 I_d)$ falling in $\mathcal{C}_1$ and $\mathcal{C}_0$ respectively is $F_{\chi_d^2}(r)$, where $F_{\chi_d^2}$ is the CDF of a $\chi_d^2$ random scalar. By picking $r = d + \sqrt{2dt} + 2t$ for some $t > 0$, the probability of $x_1 \in \mathcal{C}_1$ and $x_0 \in \mathcal{C}_0$ is at least $(1 - e^{-t})^2$. $\qquad\square$

## 5   Conclusion

We studied the problem of support data sensitivity in meta-learning: the performance of existing algorithms is extremely sensitive to the examples used for adaptation. Our findings suggest that when deploying meta-learning, especially in safety-critical applications such as autonomous driving or medical imaging, practitioners should carefully check the support examples they label for the meta-learner. However, even when the data is interpretable for a human, e.g. images, recognizing potentially detrimental examples could be hard as we have seen in our experiments.

In our experiments, we considered popular few-shot image classification benchmarks. We note that meta-learning has also been applied to data in other modalities such as language understanding [9] and speech recognition [16]. We expect our conclusions and Algorithm 1 for finding the worst-case support data to apply in other meta-learning applications, however, an empirical study is needed to verify this.

Going forward, our results suggest that robustness in meta-learning could be achieved by explicitly encouraging separation and tighter intra-class embeddings (at least in the context of last-layer adaptation meta-learners). Unfortunately, the adversarial training approach, while successful in promoting robustness in many applications, fails to achieve robustness of meta-learners to support data. In our experiments, adversarial training achieved well-separated and tight intra-class embeddings resulting in robustness on the train tasks (i.e., tasks composed of classes seen during training), but failed to improve on the test tasks. Our findings demonstrate that new approaches are needed to achieve robustness in meta-learning.

Finally, we note that our results provide a new perspective on a different meta-learning phenomenon studied in prior work. Setlur et al. [36] and Ni et al. [27] studied the importance of the support data during training. Setlur et al. [36] quantified the impact of the support data diversity, while Ni et al. [27] considered the effectiveness of the support data augmentation, and both concluded that meta-learning is insensitive to the support data quality and diversity. In our experiments with a variation of adversarial training in Section 3.3, where support data during training is selected based on Algorithm 1 to find worst-case support examples, we achieved significant improvements in terms of the worst-case accuracies on the train tasks. Thus, the conclusion is different from [36, 27], i.e. the support data used during training has an impact on the resulting meta-learner from the perspective of sensitivity studied in our work.

## Acknowledgments and Disclosure of Funding

This note is based upon work supported by the National Science Foundation (NSF) under grants no. 1916271, 2027737, and 2113373. Any opinions, findings, and conclusions or recommendations expressed in this note are those of the authors and do not necessarily reflect the views of the NSF.

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
