# A   Experiment details

**Network architectures:**   For MAML and Meta-Curvature experiments, we use a 4-layer CNN network, where each convolutional block in the network is a sequential composition of a [$2 \times 2$ max-pooling layer, batch normalization, and a $3 \times 3$ convolution layer]. The final classification layer of the network is fully-connected layer mapping the input features to 5-way output.

For ProtoNet and R2D2 experiments, we use the same architectures as are used in the original papers. The ProtoNet feature extractor is a combination of four convolutional blocks. Each block consists of a 64-filter $3 \times 3$ convolution, batch normalization layer, a ReLU nonlinearity, and a $2 \times 2$ max-pooling layer. The R2D2 feature extractor is a combination of 4 convolutional layers with [96, 192, 384, 512] filters. Each convolutional layer consists of a $3 \times 3$ convolution, batch normalization, $2 \times 2$ max pooling, and a leaky ReLU with a factor of 0.1

For MetaOptNet experiments, we use the same implementation and setting as described in the original paper. MetaOptNet networks consist of a ResNet-12 network as feature extractors, and either a support vector machine or ridge regression based head for classification.

**Meta-learning setup:**   MetaOptNet networks utilize SGD with Nesterov momentum of 0.9 and a weight decay of $5 \times 10^{-4}$ for optimization. The learning rate for this set of experiments was initially set to 1.0 and then modified to 0.06 for epochs 20 to 40, 0.012 for epochs 40 to 50, and 0.024 thereafter. MAML and Meta-Curvature networks are trained using Adam optimizer with an initial learning rate of $3 \times 10^{-4}$ and 0.01 respectively.

All networks are trained for 60000 iterations – 60 epochs of 1000 episodes each, with the batch sizes for MAML experiments set to 32 tasks in each batch, batch size for Meta-Curvature set to 16 tasks in each batch, and for MetaOptNet experiments it's set to 8 tasks in each batch.

During the meta-training phase, we apply the random crop, color jitter, and random horizontal flip transformations for MetaOptNet networks. Additionally, we match the meta-training shot with the meta-testing shot for all networks. While meta-training, we compute the accuracy on a 5-shot 5-way validation dataset, and select the model with the best accuracy on this validation dataset for sensitivity analysis.

During evaluation, i.e. results in Tables 1, 2, and 3, to compute best and worst accuracies we randomly partition each class in each task into 400 potential adaptation examples composing $\mathcal{X}$ and 200 evaluation examples composing $\mathcal{D}$ (all datasets have 600 examples per class). We use the corresponding algorithm to find the adaptation examples in $\mathcal{X}$ and report mean and standard deviations of evaluation data $\mathcal{D}$ post-adaptation accuracies over 500 random tasks for 1-shot setting, and 100 random tasks for 5-shot and 10-shot settings. To compute average accuracies we follow the setup of previous meta-learning papers, i.e. sample corresponding number of adaptation examples randomly and choose a random subset of 50 examples per class for evaluation, and report mean and standard deviation of accuracies over 1000 random tasks.

**Adversarial training setup:**   To adversarially train the models as described in Section 3.3, we initialize with models trained in the standard fashion. We then train these models in an adversarial manner for 60 epochs of 1000 episodes each. We use the same hyperparameters and experimental setup as we did for the standard training, except that we reduce the learning rate by a factor of 10. Thus, the learning rate is initially set to 0.1 for the first 20 epochs, then modified to 0.006 for epochs 20 to 40, 0.0012 for epochs 40 to 50, and 0.0024 thereafter. To find the adversarial examples, we find the worst-case examples using algorithm 1 run for 3 iterations. We then use these worst-case examples as the query data to update the model parameters.

**Algorithm runtimes:**   We run all our experiments on a 12 CPU core, 32 GB RAM, and 1 V100 GPU machine. The run times for a single iteration (in our experiments we ran 3 iterations to find the worst/best case examples) for MetaOptNet-SVM method on CIFAR-FS are approximately 3 minutes for 1-shot setting, approximately 18 minutes for 5-shot setting, and approximately 43 minutes for 10-shot setting. The corresponding run times for a single iteration for R2D2 method on FC100 are approximately 1 minute for 1-shot setting, approximately 6 minutes for 5-shot setting, and approximately 20 minutes for 10-shot setting.

# B Convergence of the algorithm to find adaptation vulnerabilities

To find the worst-case support examples (Section 3) and to find the adversarial examples for adversarial training (Section 3.3), we use Algorithm 1 with 3 rounds of attack or iterations. In this section, we show the convergence of Algorithm 1 and the rationale behind choosing 3 iterations. We execute algorithm 1 to find the worst-case 5-way 10-shot support examples on CIFAR-FS and FC100 datasets for R2D2, ResNet-Ridge, and the ResNet-SVM algorithms. We track the worst-case accuracy as it updates through 10 iterations, and show the mean worst-case accuracy through the iterations averaged over 5 different randomly-sampled tasks in figure 6. We see that while the worst-case accuracy drops significantly in the first few iterations, it stabilizes after 3 iterations and does not show significant change after the 3 iterations for all datasets and algorithms. Additionally, running for longer iterations can reduce accuracy slightly more for the 10-shot setting as compared to the 1-shot and 5-shot settings; however, the 10-shot setting is more robust to this algorithm than the 5-shot setting and thus the drop in accuracy in later iterations is understandable.

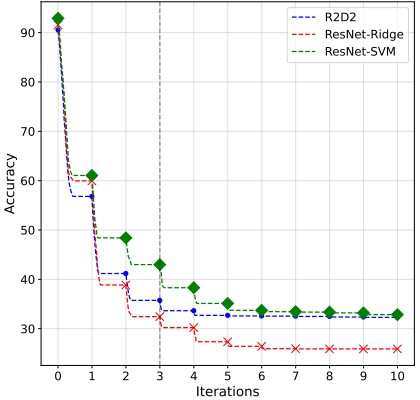
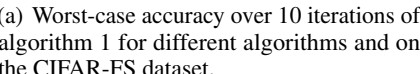
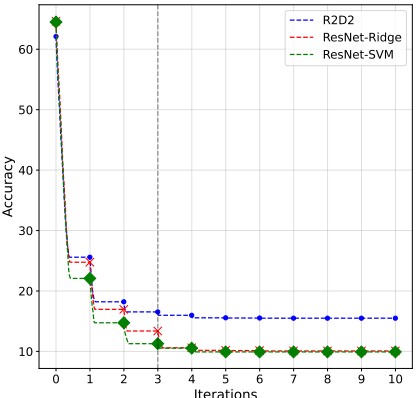

(a) Worst-case accuracy over 10 iterations of algorithm 1 for different algorithms and on the CIFAR-FS dataset.

(b) Worst-case accuracy over 10 iterations of Algorithm 1 for different algorithms and on the FC100 dataset.

Figure 6: Convergence plots for Algorithm 1 for different meta-learning algorithms on the CIFAR-FS and FC100 datasets.

# C Performance range results

1. In Figure 7 we present several examples of the worst-case support images on the FC100 and the miniImageNet datasets for the MetaOptNet-SVM method. Additionally, in Figure 8 we present several examples of the worst-case support images on the CIFAR-FS, FC100, and the miniImageNet datasets for the ProtoNets method. All the support images are correctly labeled and appear representative of the respective classes. Closely inspecting the images, we note that it is often not easy to notice visually that such support examples could result in a poor performance without significant expert knowledge of the dataset. Barring a very small portion of the images (e.g., gray-scale "Girl" image and a truck-size "Nematode"), images appear reasonable and adequately representative of their respective classes.

2. In Figure 9 we present histograms of accuracies visualizing the first iteration over classes of Algorithm 1 in 1-shot learning on miniImageNet dataset for MAML, R2D2, MetaOptNet-Ridge, and the MetaOptNet-SVM algorithms. The rightmost histogram (in each sub-figure) corresponds to post-adaptation accuracies for different choices of support image for class 0 and random choices for classes 1-4. The subsequent histogram (in each sub-figure) is for different choices of support images for class 1, where image for class 0 is chosen with Algorithm 1 and classes 2-4 are random, and analogously for the remaining three histograms. We see that the lower accuracy tails of the first two histograms contain multiple worst-case support examples in the corresponding classes. By the third histogram, the range of accuracies is well below the average accuracies for each of the algorithms for *all* possible support examples in the corresponding classes.

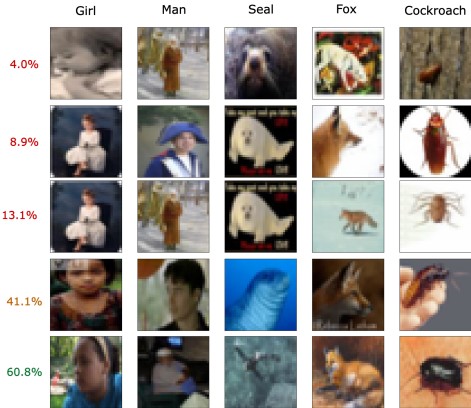
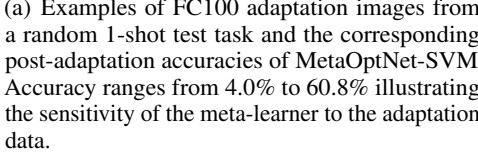
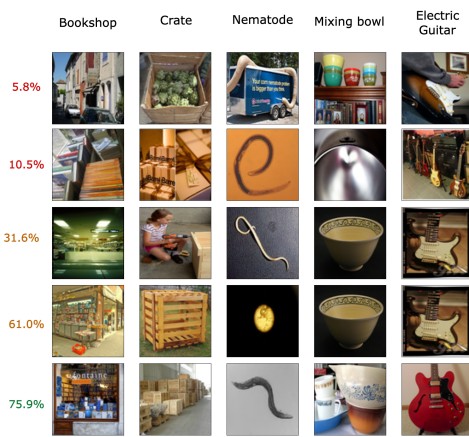

(a) Examples of FC100 adaptation images from a random 1-shot test task and the corresponding post-adaptation accuracies of MetaOptNet-SVM. Accuracy ranges from 4.0% to 60.8% illustrating the sensitivity of the meta-learner to the adaptation data.

(b) Examples of MiniImageNet adaptation images from a random 1-shot test task and the corresponding post-adaptation accuracies of MetaOptNet-SVM. Accuracy ranges from 5.8% to 75.9% illustrating the sensitivity of the meta-learner to the adaptation data.

Figure 7: Examples of unaltered support images from the FC100 and miniImageNet datasets for a random 1-shot task, depicting the post-adaptation performance of a popular meta-learning algorithm (MetaOptNet-SVM).

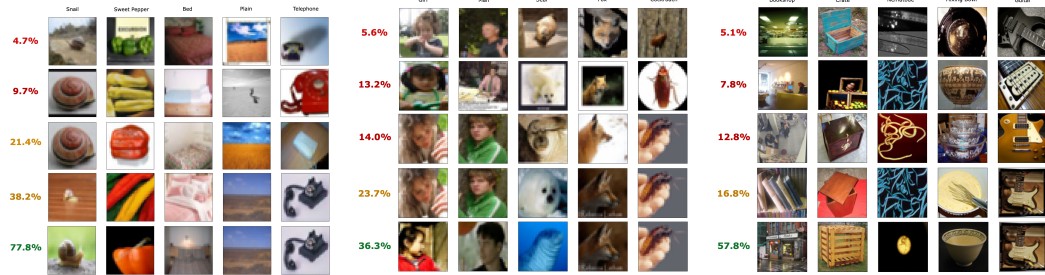

(a) Examples of CIFAR-FS adaptation images from a random 1-shot test task and the corresponding post-adaptation accuracies of ProtoNet.

(b) Examples of FC100 adaptation images from a random 1-shot test task and the corresponding post-adaptation accuracies of ProtoNet.

(c) Examples of MiniImageNet adaptation images from a random 1-shot test task and the corresponding post-adaptation accuracies of ProtoNet.

Figure 8: Examples of unaltered support images from the CIFAR-FS, FC100, and miniImageNet datasets for a random 1-shot task, depicting the post-adaptation performance of a popular meta-learning algorithm (ProtoNets).

3. In Figure 10 we present histogram of accuracies of unique combinations from the miniImageNet dataset of 1-shot support examples evaluated by Algorithm 1 throughout 3 iterations.

   (a) For the MAML algorithm, out of the 4390 distinct sets represented in the histogram, 3054 distinct sets of 5 examples each have less than 30% post-adaptation accuracy.

   (b) For the R2D2 algorithm, out of the 5986 distinct sets represented in the histogram, 5274 distinct sets of 5 examples each have less than 40% post-adaptation accuracy.

   (c) For the MetaOptNet-Ridge algorithm, out of the 3592 distinct sets represented in the histogram, 2795 distinct sets of 5 examples each have less than 40% post-adaptation accuracy.

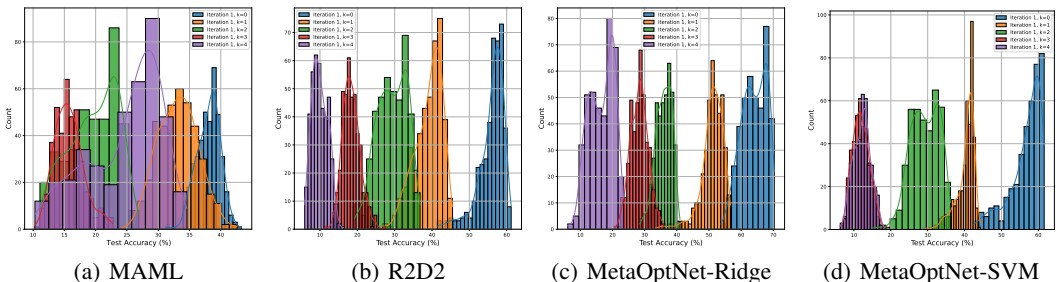

| (a) MAML | (b) R2D2 | (c) MetaOptNet-Ridge | (d) MetaOptNet-SVM |

Figure 9: Histogram of accuracies visualizing progression of the first iteration of Algorithm 1 in 1-shot learning over the miniImageNet dataset for different meta-learning algorithms.

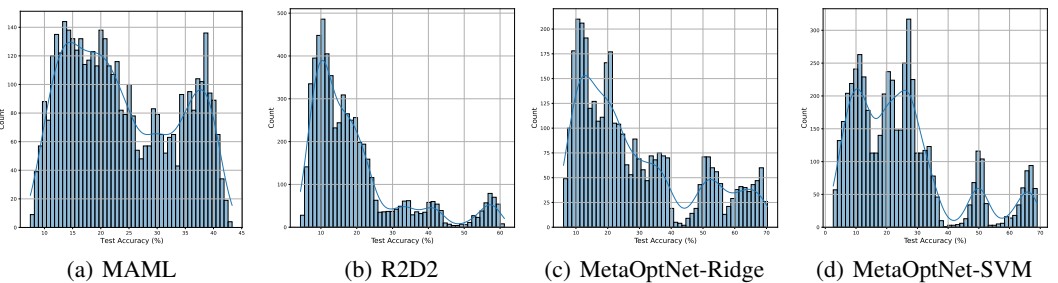

| (a) MAML | (b) R2D2 | (c) MetaOptNet-Ridge | (d) MetaOptNet-SVM |

Figure 10: Histogram of accuracies of unique combinations from the miniImageNet dataset of 1-shot support examples evaluated by Algorithm 1 for 3 iterations,

(d) For the MetaOptNet-SVM algorithm, out of the 5188 distinct sets represented in the histogram, 4403 distinct sets of 5 examples each have less than 40% post-adaptation accuracy.

# D   Improving support data robustness with adversarial training

In Table 6, we show the projected embeddings for the R2D2, MetaOptNet-Ridge, and the MetaOptNet-SVM algorithms on the training and the test dataset, when trained in a standard manner vs when trained adversarially. As we note in Section 3.3 and as results depict in Tables 4 and 5, the adversarial training converges and the worst-case accuracy improves drastically on the training tasks while no improvement is observed on the test tasks.

Table 6: Projected embeddings of R2D2, MetaOptNet-Ridge, and MetaOptNet-SVM methods for a train and a test task query data from the FC100 dataset. We compare standard training and adversarial training discussed in Section 3.3. Points are colored with their labels. Highlighted points are the worst-case support examples selected with Algorithm 1.

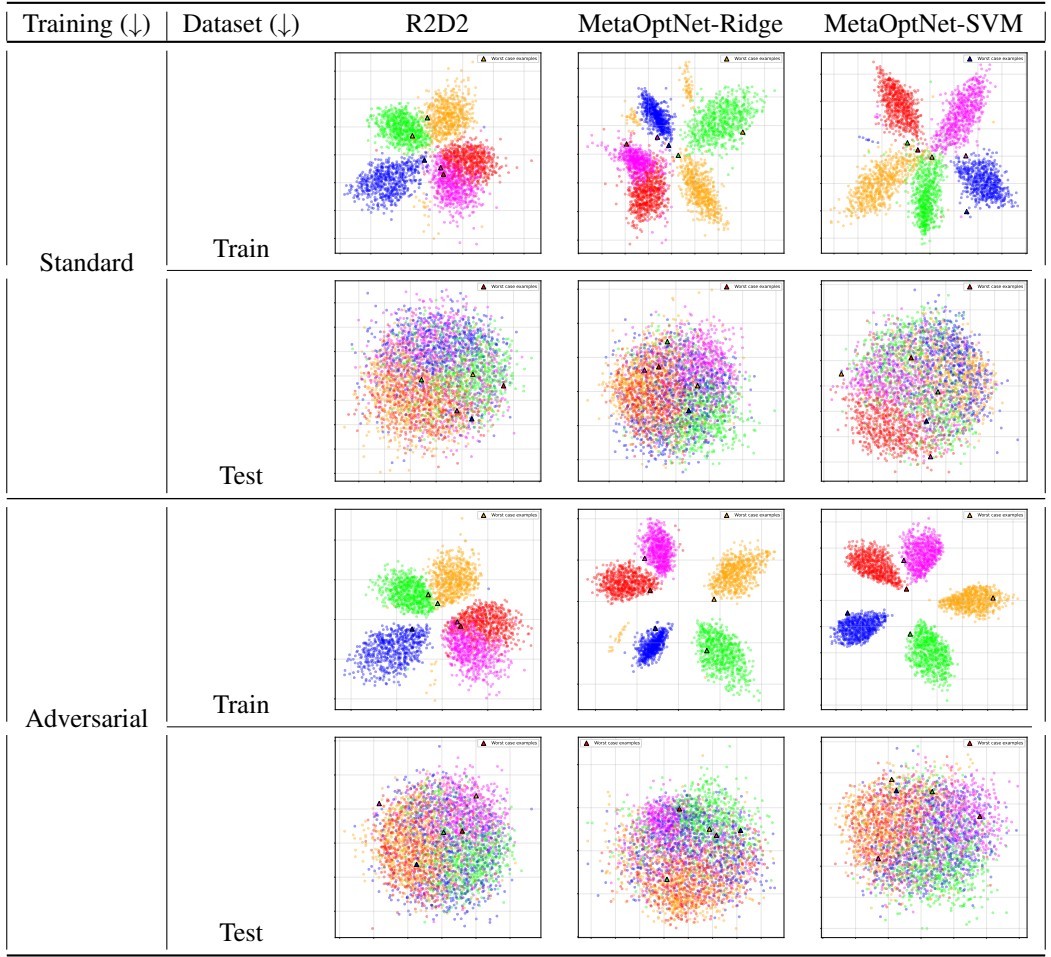