# OpenReview forum: "On sensitivity of meta-learning to support data"
_NeurIPS.cc/2021/Conference — NeurIPS 2021 Poster_

### Official Review · Reviewer_kYmb · 2021-06-25

**Rating:** 7
**Confidence:** 4

**Summary:**

This work observes that meta-learners are incredibly sensitive to support data for fine-tuning and explains this phenomenon in terms of margins.

**Limitations And Societal Impact:**

The authors do adequately address the limitations and impact of their work.

**Main Review:**

1)  Better understanding exactly how meta-learning works and understanding its properties is an interesting direction and is important given the popularity and success of meta-learning.

2)  [1] examines sensitivity to support data (during training) from an adversarial view in which the worst-case support data is not natural images but adversarially perturbed ones.  It would be good to relate your work to the findings from this paper.

3)  [2] studies the sensitivity of meta-learning to the support set during training rather than testing in Section 3.  This work should also be considered in your paper.

[1] Attacking Few-Shot Classifiers with Adversarial Support Sets
[2] Data Augmentation for Meta-Learning

If the authors relate their work to the above papers and justify the novelty and significance of their conclusions in the context of these works, I am open to raising my score.

[Update: increased score to reflect that my comments were addressed]

**Time Spent Reviewing:**

3

---

> ### Author Response · Authors · 2021-08-11
> **Response**
>
> We thank the reviewer for the feedback and for the related work pointers. We discuss them below as you requested.
>
> **[1] Attacking Few-Shot Classifiers with Adversarial Support Sets**
>
> [1] proposes an adversarial attack on the support set that is used for adaptation to a test task (citing [1], "the attack is perpetrated at meta-test time, after the meta-learner has already been meta-trained"). The key difference between [1] and our work, as you noted, is that we consider an adversary-free setting. That is, we show that even when meta-learning is applied in a setting where an adversary intervention is unlikely, meta-learning may still fail after adapting on support sets consisting of natural, unaltered images. We have discussed prior works considering various adversarial attacks in lines 27-34 and cited another paper that studied adversarial attacks on the support set. We will cite [1] in the revised version. To comment on the novelty, adversarial vulnerabilities of meta-learning (and deep learning more broadly) have been studied in many prior works, while failures in adversary-free settings have not been previously studied in meta-learning to the best of our knowledge. On the significance of our findings, while adversarial vulnerabilities is an important problem in deep learning, failures in adversary-free settings are even more alarming. We hope that our findings will serve as a ground for evaluating meta-learners sensitivity in adversary-free settings, and for developing new meta-learning methods that are more robust.
>
> **[2] Data Augmentation for Meta-Learning**
>
> [2] studies effectiveness of a variety of data augmentation strategies applied to different meta-learning stages. They mostly advocate for query augmentation during training. In regards to the support data augmentation during training they state "meta-learning is fairly insensitive to the amount and quality of support". In our work we study sensitivity to support data during evaluation and arrive to a different conclusion regarding the importance of support data during training. Specifically, in section 3.3 we study a variation of "adversarial" training, where support data during training is selected based on Algorithm 1 to find worst-case support examples (i.e. without doing any modifications to the images). In Tables 4 and 5 we show that training with worst-case support data greatly improves robustness of meta-learning to the support data when evaluated on tasks composed of classes seen during training (see also Figure 4 (a) and (c) for a visualization of the improvement in embeddings), however fails to improve the worst-case performance on the test tasks composed of unseen classes. We will extend the discussion in section 3.3 to compare our conclusions regarding the importance of support data during training to [2].

---

> ### Author Response · Authors · 2021-08-25
> **Discussion with Reviewer kYmb**
>
> Dear Reviewer kYmb,
>
> We thank you again for pointing us to the two related works. In our response, we compare our work to the two related works and clarify the novelty and significance of our work. Please let us know if you have any other questions regarding our contributions. We also ask that you consider increasing your score if you are satisfied with our response.

---

> > ### Comment · Reviewer_kYmb · 2021-08-30
> > **Thank you for your response**
> >
> > Thank you for your response and for adding discussions of these relevant works.  I have increased my score.

---

### Official Review · Reviewer_Vkec · 2021-07-12

**Rating:** 7
**Confidence:** 4

**Summary:**

This paper is studying the performance of several meta-learning approaches depending on the support images used for meta-test tasks.
First, the authors propose a simple greedy algorithm that select for each class in a given task, the image giving the worst/ the best performance. With this algorithm, they show that popular meta-learning approaches have a very high variance in their performance and that the worst case scenario is often worse than a random guess. The authors argue that the worst tasks found by their algorithm are not artifacts in the distribution, and can realistically appear.
Then, they apply their algorithm to train the meta-learning methods in an adversarial way, by learning on the worst cases found by their algorithm. Even though the adversarial training improves the worst-case performance on the meta-training tasks, there are no improvements on the worst-case accuracy on meta-testing tasks.
Finally, the authors argue that the worst-case performance are better when the task embeddings are sufficiently separated.

**Ethical Concerns:**

I do not see any ethical issues with this paper.

**Limitations And Societal Impact:**

The authors did not discuss potential negative impacts of their work. Even though the work presented here is not fully theoretical, it is high upstream of the applications. To study the possible negative impacts of this kind of work is to study the negative impacts of the whole field, since it is discussing the way we are evaluating meta-learning models. As any other field in machine learning, meta-learning could also be used for harmful applications, such as quick racial profiling of a user.

**Main Review:**

__**Originality:**__  The paper presents an interesting approach to find worst/best case performance and a novel evaluation of performance for meta-learning methods. The evaluations are similar to some experiments in semi-supervised setting [1], but it was never investigated in the case of meta-learning.
 This paper shows actual evidence that there is a high variance when evaluating meta-learners. It is the first time I see an actual computation of worst and best performance, even though the problem was already raised [2].

__**Quality:**__  The paper propose a simple but effective greedy algorithm for computing worst/best case performance. If I understand correctly, the algorithm searches from *all* possible support images for a given class. I think it would be interesting to have the computing time of this algorithm, since it seems quite heavy to compute the performance for all possible images.
Using an adversarial training scheme with the proposed algorithm is interesting, even though it did not help to improve the worst case performance. It shows that reducing the variance in performance is complicated.

__**Clarity:**__  The main message and the writing of the paper are clear. The paper is also well-structured.
However, I think the discussions about the experiments could be improved, particularly section 3.2:
- Each point in the section could benefit from their own conclusion to link it to the main point that worst support examples are not artifacts. Currently, the reader must draw the conclusion from each point themselves.
- Figures 2 and 3 are hard to understand because they appear before their introductions and the explanations of the experiments in the text. This makes it difficult to understand the conclusions that we have to draw from them.
- The meaning of an *iteration* of the algorithm should be clarified. If I understand correctly, an iteration means going through all support images for each class. I had to look at the code to really understand the algorithm.

__**Significance:**__ I think the paper is valuable for practitioners. It brings concrete evidence that there is high variance in performance of meta-learning methods. This is currently a significant problem in few-shot learning. The adversarial training experiment is a naive idea to try to alleviate this problem. Even though the authors show that it doesn't work, it could help future work.

__**Minor Remarks:**__

- What projection method was used ? I assumed it was t-SNE.
- Experimental details for ProtoNets and R2D2 are missing in the appendix. From the performance, I assumed the authors used a ResNet-12 backbone, similarly to MetaOptNet.
- Why is there so much variance for MAML and MC on average accuracy ? The variance is a lot higher than for other methods, and I never saw that much variance for these algorithms. I assumed it was a typo when reporting results.

[1]: Sohn et al., *FixMatch: Simplifying Semi-Supervised Learning with Consistency and Confidence.* NeurIPS 2020
[2]: Dhillon et al., *A baseline for few-shot image classification.* ICLR 2020

**Time Spent Reviewing:**

I spent about 5 - 6 hours to review this paper.

---

> ### Author Response · Authors · 2021-08-11
> **Response**
>
> We thank you for the feedback. Below we answer the questions raised in the review.
>
> **Run time of Algorithm 1**
>
> As the reviewer noted, Algorithm 1 is quite intensive computationally. We note that its purpose is to demonstrate the problem, therefore the run-time is less of an issue (as opposed to a method for solving a problem, that need to be computationally efficient to be practical). The run-times of a single iteration (in our experiments we ran 3 iterations to find the worst/best case examples) for MetaOptNet-SVM on CIFAR-FS are approximately 3 minutes for 1-shot and approximately 18 minutes for 5-shot settings on a 12 CPU core, and 1 V100 GPU machine. We will add additional run-time results in the revised version.
>
> **Comments regarding clarity**
>
> We thank the reviewer for the positive feedback regarding the overall clarity of the paper. We will incorporate the suggested improvements in Section 3.2. The reviewer understood the meaning of an iteration correctly and we will clarify it in the text. We also appreciate the reviewers effort in looking at the code.
>
> **What projection method was used?**
>
> We used Multidimensional Scaling (MDS) to preserve the relative cluster sizes and distances between clusters. Our theoretical analysis emphasizes the importance of class margins and MDS can visualize them better since its goal is to approximate the distances between data points (tSNE tends to spread the clusters further apart, thus it may give a false idea of the class margins). We will clarify this point in the revision.
>
> **Experimental details for ProtoNet and R2D2 missing. From the performance, I assumed the authors used a ResNet-12 backbone, similarly to MetaOptNet.**
>
> We use the same architectures as was proposed in the original ProtoNet [1] and R2D2 [2] papers. We will add details for these in the appendix:
> - The ProtoNet feature extractor is a combination of four convolutional blocks. Each block consists of a 64-filter 3 × 3 convolution, batch normalization layer, a ReLU nonlinearity and a 2 × 2 max-pooling layer.
> - The R2D2 feature extractor is a combination of 4 convolutional layers with [96, 192, 384, 512] filters. Each convolutional layer consists of a 3 × 3 convolution, batch normalization, 2 × 2 max pooling, and a leaky ReLU with a factor of 0.1
>
> **MAML variance**
>
> We revisited the experiments to check for high variances for MAML and MC, and there were 2 key reasons behind them:
> - The evaluation setup for the *average* accuracy of MAML and MC differed from the remaining algorithms. While the other algorithms were tested on 50 query samples per class, MAML and MC were tested on 1 query sample per class (this was a bug in the code). We emphasize that this was only the case for the *average* accuracy, worst/best accuracies were reported correctly. Because of the small size of the query data, the variance was high. We've recomputed the average accuracies for MAML and MC using 50 query samples per class, and while the mean accuracy has not changed much, the standard deviation has reduced considerably. We've updated these results in the paper, and are also providing the results for MAML in the table below for your reference.
> - Besides the difference in the evaluation setup, we would also like to note that other papers such as MAML [3], MC [4], ProtoNets [1], R2D2 [2], and MetaOptNet [5], all report the 95% confidence interval for accuracy while we report the standard deviation. We computed the 95% confidence interval along with the mean and the std of the accuracy, and found that the numbers are comparable to the ones reported earlier.
>
> **Updated MAML average accuracy results:**
>
> |              |         | Mean accuracy | Std | CI-95 |
> |:------------:|:-------:|:----------:|:---------:|:----------:|
> |              |  1-shot |    56.52   |   10.85   |    0.95    |
> |   CIFAR-FS   |  5-shot |    70.45   |    8.42   |    0.74    |
> |              | 10-shot |    70.88   |    7.82   |    0.68    |
> |              |  1-shot |    31.89   |    6.75   |    0.59    |
> |     FC100    |  5-shot |    43.58   |    7.61   |    0.67    |
> |              | 10-shot |    44.30   |    6.89   |    0.60    |
> |              |  1-shot |    47.13   |    8.78   |    0.77    |
> | miniImageNet |  5-shot |    57.69   |    7.92   |    0.69    |
> |              | 10-shot |    59.52   |    8.34   |    0.73    |
>
>
> **Potential negative impacts.**
>
> We agree with the reviewer that "As any other field in machine learning, meta-learning could also be used for harmful applications" and will add a corresponding discussion to the conclusion.
>
> **Additional related work.**
>
> Thank you for the related work pointers. We will discuss them in the revised version.
>
> ---
>
> [1] Jake Snell, Kevin Swersky, and Richard S Zemel, 2017. Prototypical networks for few-shot learning.
>
> [2] Luca Bertinetto, Joao F Henriques, Philip Torr, and Andrea Vedaldi, 2019. Meta-learning with differentiable closed-form solvers.
>
> [3] Chelsea Finn, Pieter Abbeel, and Sergey Levine, 2017. Model-agnostic meta-learning for fast adaptation of deep networks.
>
> [4] Eunbyung Park and Junier B Oliva, 2019. Meta-curvature.
>
> [5] Kwonjoon Lee, Subhransu Maji, Avinash Ravichandran, and Stefano Soatto, 2019. Meta-learning with differentiable convex optimization.

---

> > ### Comment · Reviewer_Vkec · 2021-08-26
> > **Thanks for the clarifications - Updated score**
> >
> > I thank the authors for their detailed response, they addressed all my remarks.
> > I appreciate that the authors have recomputed the variance/CI for the average case.
> > About the running time of Algorithm 1, I agree that it is a lesser issue but I think it is interesting to have the order of magnitude of its running time.
> > With these clarifications taken into account, I raise my score to 7.

---

> > > ### Author Response · Authors · 2021-08-26
> > > **Thank you**
> > >
> > > We thank the reviewer for helping us improve the paper. We will incorporate your suggestions (including run-time analysis for Algorithm 1) in the final version.

---

### Official Review · Reviewer_Yniy · 2021-07-15

**Rating:** 6
**Confidence:** 3

**Summary:**

The paper studies the sensitivity of meta-learning to the support data. The paper presents an algorithm for finding the best and worst support examples relative to the test accuracy, and demonstrates empirically the sensitivity of several meta-learning methods to the support data. The paper explored using  adversarial robustness strategies, however, while these methods seem to work on the training tasks, they fail to generalize on new test tasks. The paper explores the sensitivity of meta-learning algorithm to the support data from the margin perspective, arguing that higher margins and interclass tightness leads to more robust meta-learning methods.

**Limitations And Societal Impact:**

- Empirical analysis was only conducted in a few-shot text classification setup.
- Actionable solutions for mitigating the sensitivity of the meta-learning methods to support data wasn’t presented.

**Main Review:**

## Originality

- The paper presents a greedy search algorithm (Algorithm 1) to study the sensitivity of meta-learning algorithms to the support data. The empirical analysis presented in the paper to quantify this sensitivity is novel and useful for the meta-learning practitioners to be informed about.However, it’s limited only to few-shot classification tasks, and doesn’t include other meta-learning applications  for instance in language and speech.

- Related work was discussed in the introduction and background section, some missing references include: “ Setlur, Amrith, Oscar Li, and Virginia Smith. "Is Support Set Diversity Necessary for Meta-Learning?." arXiv preprint arXiv:2011.14048 (2020).” and the references within.

## Quality

- The submission is technically sound, and the claims are well supported empirically. The paper provides a simple theoretical analysis for the effect of increasing the classification margin on robustness.

- The proposed greedy search method is appropriate and succeeds in finding the best and worst support training point, despite lacking the guarantees for global convergence.

- Evaluation was performed only on computer vision tasks.

## Clarity

- The submission is clearly written, well organized, and easy to follow.
- Algorithm 1 could be illustrated better with a figure or a running example.

## Significance

- The analysis is useful for the meta-learning community to be informed about, however, no direct solution for the problem was presented in the paper. The paper presents the insight that maximizing the classification margin and minimizing inter-class variations would improve robustness. Adversarial training didn’t improve the results significantly on new test tasks.

**Time Spent Reviewing:**

3

---

> ### Author Response · Authors · 2021-08-11
> **Response**
>
> We thank you for the review. Please see our responses below.
>
> **Discussion of the additional reference.**
>
> Thank you for bringing up the work of [1]! It is an interesting paper in the context of our work. [1] studies the importance of the support data diversity during training and concludes that it is not an important aspect of meta-learning. In our work we majorly focus on the sensitivity to support data during evaluation, however our experiments in Section 3.3 also reveal some insights regarding the importance of support data during training. Specifically, in Section 3.3 we study a variation of "adversarial" training, where the support data during training is selected based on Algorithm 1 to find worst-case support examples. In Tables 4 and 5 we show that training with worst-case support data greatly improves robustness of meta-learning to the support data when evaluated on tasks composed of classes seen during training (see also Figure 4 (a) and (c) for a visualization of the improvement in embeddings), however fails to improve the worst-case performance on the test tasks composed of unseen classes. Thus the conclusion is different from [1], i.e. the support data used during training has an impact on the resulting meta-learner from the perspective of sensitivity studied in our work. We will extend the discussion in section 3.3 to compare our conclusions regarding the role of the support data during training to [1].
>
> **It’s limited only to few-shot classification tasks, and doesn’t include other meta-learning applications for instance in language and speech.**
>
> We acknowledge that our evaluation is limited to few-shot image classification task, and does not look at language and speech applications along with many other applications we might have missed. With the many applications of meta-learning, benchmark datasets, and methods, it is hardly possible for a single paper to consider all of them. In this work we provide and extensive study of the problem in the context of few-shot learning on images. We expect our conclusions and method for finding the worst case support data to apply in other meta-learning applications, however an empirical study is needed to verify this. In the revised version we will mention the limitation of our study in terms of application breadth, as you suggested.
>
> **Algorithm 1 could be illustrated better with a figure or a running example.**
>
> Thank you for the suggestion. Figure 2 partially serves this goal, but we will add a more visual illustration of the algorithm by showing the specific worst-case images throughout its progression.
>
> **Actionable solutions for mitigating the sensitivity of the meta-learning methods to support data wasn’t presented.**
>
> We show that standard approaches to robustifying ML models do not address the sensitivity of meta-learning method (see section 3.3). Thus we need new representation-learning approaches to solve this issue. This is beyond the scope of the paper.
>
> [1] Setlur, A., Li, O. and Smith, V., 2020. Is Support Set Diversity Necessary for Meta-Learning?

---

> > ### Comment · Reviewer_Yniy · 2021-08-30
> > **Thanks for the Clarifications**
> >
> > I thank the authors for their detailed response and clarification. It’d be great to include the clarification above to the related work section of the paper, and add the limitations and future study for other ML tasks in NLP and Speech as future work items. After reading the authors’ response, I think the paper is above the acceptance threshold.

---

> > > ### Author Response · Authors · 2021-08-30
> > > **Thank you**
> > >
> > > We thank the reviewer for helping us improve the paper. We will incorporate the suggested changes in the final version.

---

### Official Review · Reviewer_fbrp · 2021-07-16

**Rating:** 6
**Confidence:** 5

**Summary:**

This paper empirically finds that popular meta-learning algorithms are very sensitive to the selections of the support images. The author shows that six meta-learning algorithms can achieve testing accuracy within a wide accuracy (worst case accuracy is extremely poor) given different support samples. They also find that this problem cannot be trivially solved by adversarial training.


**Limitations And Societal Impact:**

1) I wonder whether this sensitivity to support data is only to meta-learning or to the whole few-shot learning. For example, will the pre-trained feature embedding from regular supervised training or self-supervised learning suffer from the same problem?

2) I think we need more analysis for the worst cases. For example, are the worst cases the same across different meta-learners?  How do the feature representations perform for the worst case images, are they on the boundary of classification?

3) I am not sure how important the question is, especially for 1-shot cases. As long as your model is not 100% accurate for meta-test tasks, you can always find the misclassification cases, and set them to be support data in which case I don’t think it’s possible to get a good accuracy.

**Main Review:**

The paper is easy to follow and the topic and findings are interesting to me. The robustness to the support data seems important to areas such as medical data and autonomous driving. It’s interesting to see the robustness of large support samples (10-shot) is not robust enough to testing samples. However, the authors fail to get a solution to this problem and thus their analysis part is not enough to address how important the problem is.


**Time Spent Reviewing:**

3

---

> ### Author Response · Authors · 2021-08-11
> **Response**
>
> We thank the reviewer for the feedback. Please find our responses below.
>
> **Will the pre-trained feature embedding from regular supervised training or self-supervised learning suffer from the same problem?**
>
> It depends on the quality of the learned representations as shown in Theorem 4.1 and discussed in Section 4 (those results apply to any method for extracting embeddings). Our work demonstrates the importance of learning well-separated representations (beyond linear-separability sufficient for regular supervised learning) for few-shot learning tasks. We demonstrate that the representations learned by meta-learning methods are not sufficiently separated, so the model is sensitive to the support data. The choice to study meta-learning methods over regular supervised learning is motivated by [1], where it is shown that meta-learning methods that only adapt the head of the network, e.g. MetaOptNet, learn representations that are clustered a lot better than feature representations obtained with regular supervised learning.
>
> **I think we need more analysis for the worst cases. For example, are the worst cases the same across different meta-learners?**
>
> In Section 3.2 we study worst-case examples from four different perspectives (items 1-4). For items 2, 3, and 4 we present results for a wide variety of meta-learners (see main text and supplement Section C). Thus, we present an extensive study of the worst case examples across variety of meta-learners. For item 1 we instead focused on studying different datasets for a single meta-learner, i.e. MetaOptNet-SVM (see Figures 1 and 6). Per your suggestion, we will also add worst-case image examples for other meta-learners in the supplement of the revised version.
>
> **How do the feature representations perform for the worst case images, are they on the boundary of classification? As long as your model is not 100% accurate for meta-test tasks, you can always find the misclassification cases, and set them to be support data in which case I don’t think it’s possible to get a good accuracy.**
>
> We assume the reviewer means true class boundaries (classification boundary is learned based on the support data). We do NOT expect all worse case images to be close to the class boundary. In fact, based on our theoretical analysis, we expect some of the worst-case images to be far from it. We provide a 2-dimensional visualization of the feature representations and the corresponding worst case images in Figure 4 to provide intuition. In (a) we show feature representations of a regular meta-learner which are linearly separable (see footnote on page 9 - the corresponding accuracy of optimal linear SVM is almost 100%), however the worst case accuracy is as low as 10% (see Table 4). In this example some of the worst-case images are near the true class boundary (e.g., red), while others (e.g., orange), are far from it. In (b) features are close to being linearly separable (see footnote on page 9), but the worst case images are now mixed with other classes. In (c), similar to (a), features are linearly separable with the worst case images also partially close to the true class boundary (e.g., orange) and partially away from it (e.g., red). However there is a key difference -- class margins are a lot bigger yielding worst-case accuracy of about 93% (see Table 4). We hope this discussion highlights one of the key insights of the paper that we think the reviewer missed: it is *not enough* to have perfect linear-separability of the representations (i.e. being "100% accurate"). A sufficiently large margin is necessary to have a model robust to support data in 1-shot setting. Please see Section 4 for more in-depth discussion.
>
> **The authors fail to get a solution to this problem and thus their analysis part is not enough to address how important the problem is.**
>
> We would like to point out that papers studying new problems that are practical, yet challenging, are important to facilitate subsequent progress in the field. Many such papers do not immediately come with a solution, and in some cases it takes several years before a working solution is found. Perhaps the most prominent example is the original paper presenting the study of adversarial vulnerability of deep neural networks [2] that was published in 2013. Since then numerous solutions have been proposed leading to the paper demonstrating failures modes of the majority of these solutions [3]. This paper received ICML 2018 best paper award. Notice that again it did not present a solution, rather reinforced the importance and challenging aspect of the problem. By now we have effective methods for achieving adversarial robustness in supervised learning, e.g. adversarial training [4], but it took nearly 5 years. More recently, there are also many examples of papers that present interesting problems without a solution, e.g. [5] that was published in last year's NeurIPS. [5] finds that most prior research on robustness focuses on synthetic image perturbations and does not adequately transfer to distributional shifts found in real data. The authors identify and bring forth the problem and conclude that distribution shifts arising in real data are currently an open research problem.
>
> Our work presents an extensive discussion, supported empirically and theoretically, of a problem that has not been studied previously and that is, citing the reviewer, "important to areas such as medical data and autonomous driving". Further, we show that the problem is challenging, i.e. it is NOT addressed by standard approaches to robustifying ML models (see section 3.3), demonstrating the need for new representation-learning approaches to mitigate support data sensitivity. With the preceding discussion and the examples of impactful prior works we hope to convince the reviewer that our contributions can be valuable to the NeurIPS community despite the lack of a method to solve the problem right away.
>
> [1] Goldblum, M., Reich, S., Fowl, L., Ni, R., Cherepanova, V. and Goldstein, T., 2020. Unraveling meta-learning: Understanding feature representations for few-shot tasks.
>
> [2] Szegedy, C., Zaremba, W., Sutskever, I., Bruna, J., Erhan, D., Goodfellow, I. and Fergus, R., 2013. Intriguing properties of neural networks.
>
> [3] Athalye, A., Carlini, N. and Wagner, D., 2018, July. Obfuscated gradients give a false sense of security: Circumventing defenses to adversarial examples.
>
> [4] Madry, A., Makelov, A., Schmidt, L., Tsipras, D. and Vladu, A., 2017. Towards deep learning models resistant to adversarial attacks.
>
> [5] Taori, R., Dave, A., Shankar, V., Carlini, N., Recht, B. and Schmidt, L., 2020. Measuring robustness to natural distribution shifts in image classification.

---

> > ### Comment · Reviewer_fbrp · 2021-09-01
> > **Thanks for the detailed response.**
> >
> > Thanks for the detailed response. I do agree that the class margins are more important and interesting for the support set robustness of few-shot learning. And after reading the response and the related section in the paper, I believe this kind of robustness issue do exist for the few-shot problems in addition to the trivial cases (especially for the meta-training samples). As a result, I would like to raise my scores to 6.

---

### Author Response · Authors · 2021-08-25
**Discussion**

Dear Reviewers,

We thank you for your comments that have helped us to improve the paper. We hope you had a chance to take a look at our responses. We kindly ask if you could please let us know whether we addressed the respective concerns and questions. We look forward to having a fruitful discussion and would appreciate if you consider increasing the score in case our responses have addressed your concerns.

---

### Decision · Program_Chairs · 2021-09-27

**Decision:**

Accept (Poster)

**Comment:**

The submission investigates the extent to which few-shot classification algorithms are sensitive to the selection of support images. It introduces a greedy algorithm for finding the worst/best support sets for a learning episode and reports that the performance of all six evaluated approaches drastically varies as a function of the support set composition. When adversarial training is employed to increase robustness, the authors report success for training episodes, but crucially these benefits do not translate to test episodes. Finally, the submission presents theory that suggests that robustness in metric-based few-shot learners could be obtained by encouraging inter-class separation and tighter intra-class clusters.

Reviewers appreciated the paper's writing clarity and found the problem setting to be interesting and of practical significance to the community. The authors provided the clarifications requested by the reviewers. Some reviewers pointed out that the authors failed to propose an effective solution, to which the authors rightfully replied that "problem-reporting" submissions can still make a substantial contribution to the field, citing Szegedy et al.'s paper on adversarial examples as evidence. Some reviewers noted that the paper's impact could be increased by considering domains beyond image classification, but ultimately found that it meets the bar for acceptance as it stands.

I therefore recommend acceptance.